# Numerical Study on Hysteretic Behaviour of Horizontal-Connection and Energy-Dissipation Structures Developed for Prefabricated Shear Walls

**Limeng Zhu [1,2]**, **Lingmao Kong [1]** and **Chunwei Zhang [1,2,*]**

[1] School of Civil Engineering, Qingdao University of Technology, Qingdao 266033, China; zhulimeng@qut.edu.cn (L.Z.); konglingmao@qut.edu.cn (L.K.)

[2] Cooperative Innovation Center of Engineering Construction and Safety in Shandong Blue Economic Zone, Qingdao University of Technology, Qingdao 266033, China

[*] Correspondence: zhangchunwei@qut.edu.cn; Tel.: +86-053285071328



**Featured Application: This developed "double-step" horizontal-connection and energy-dissipation structure could be used for prefabricated structures such as shear walls and replaceable coupling beams. In its first step, it weakly connects two adjacent shear walls and mainly dissipates the input energy. In its second step, it could strongly integrate two separate adjacent shear walls into one unit to obtain one stronger structure member to resist the collapse of the structural system.**

**Abstract:** This study proposed a developed horizontal-connection and energy-dissipation structure (HES), which could be employed for horizontal connection of prefabricated shear wall structural system. The HES consists of an external replaceable energy dissipation (ED) zone mainly for energy dissipation and an internal stiffness lifting (SL) zone for enhancing the load-bearing capacity. By the predicted displacement threshold control device, the ED zone made in bolted low-yielding steel plates could firstly dissipate the energy and can be replaced after damage, the SL zone could delay the load-bearing and the load-displacement curves of the HES would exhibit "double-step" characteristics. Detailed finite element models are established and validated in software ABAQUS. parametric analysis including aspect ratio, the shape of the steel plate in the ED zone and the displacement threshold in the SL zone, is conducted. It is found that the HES depicts high energy dissipation ability and its bearing capacity could be obtained again after the yielding of the ED zone. The optimized X-shaped steel plate in the ED zone exhibit better performance. The "double-step" design of the HES is a potential way of improving the seismic and anti-collapsing performance of prefabricated shear wall structures against large and super-large earthquakes.

**Keywords:** energy dissipation; "double-step" characteristics; stiffness lifting; seismic performance; horizontal connection; prefabricated shear wall structural systems

## 1. Introduction

### 1.1. Research Status of the Connection for the Prefabricated Shear Wall System

In order to achieve the green and sustainable development and solve the problem of environmental protection and labor shortage, it is particularly significant to develop innovative prefabricated shear wall systems appropriately employed in tall buildings and some special structures [1–3]. The in-cast shear wall system is characterized by great lateral stiffness and bearing capacity. The traditional prefabricated shear wall system is designed according to the in-cast structure standards and its seismic performance fails to meet the requirements of the current seismic design code of buildings in China.

With the application of prefabricated structural system especially in highly seismic regions, innovative design theories and systems are to be used to endow this new prefabricated system with high seismic and resilient performance. Therefore, the utilization of some resilient energy dissipation devices in structures is sensible to enhance the performance of prefabricated shear wall systems both in the large and super-large earthquakes [4,5].

The performance of connections between prefabricated shear walls has a significant influence on the seismic behavior of the prefabricated shear wall system. The present connections mainly include cast-in-place bolt connection, casing grouting connection, reserved hole slurry anchoring connection, bolt connection and post-tensioned prestressed connection [6–8]. Shemie [9] proposed a bolt connection between prefabricated panels which makes the connection between the wall panels more flexible and their ductility could be effectively utilized. Zhu et al. [10] conducted a mechanical study on horizontal and vertical joints on fabricated large slab structures, showing that horizontal seams could decrease the lateral stiffness and the shear angle of the vertical joint has a great influence on the distribution of the internal forces. Noel and Soudki [11] performed a reciprocating loading test on prefabricated shear walls and found that the bearing process of the horizontal joint could be defined as three stages which are the elastic stage before slipping, the elastoplastic stage before the damage of horizontal joint and the total slip damage. In the final stage of the slip failure, the strength will drop by 20% and the mortar will be crushed. Sun et al. [12] developed a new-type vertical joint for prefabricated wall and experimental results demonstrated that these connections were strong enough to maintain the global seismic behavior of the prefabricated wall equal to in-cast ones. Smith and Kurama [13] studied the prestressed specimens and found that their strength and initial stiffness are similar to those of cast-in-place specimens. The test piece demonstrated slight damage with a large nonlinear displacement, good self-centering ability but a little decrease in energy dissipation ability. Vaghei et al. [14,15] tested the U-shaped steel channel wall-to-wall connection and found that this type of connection performed better than loop connections. Guo et al. [16] proposed bolt connections for prefabricated wallboard structures and conducted shaking table tests on a 1/2 scale three-story model. The results show that the prefabricated structure system has the characteristics of high stiffness, large bearing capacity, and high collapse margin ratio. Because the current design code regards grouting pile connection as an idealized steel bar, it ignores the restriction of sleeve and composite behavior of components. Son et al. [17] proposed that the sheer force of horizontal connections of members can be resisted by overlapping anchors. The shear behavior of overlapping anchors between prefabricated concrete slabs and reinforced concrete members in simulated tests is analyzed. The results show that the average shear strength of overlapping anchorage connections is 109% of the calculated value. Jiang et al. [18] studied the effect of new bolted connections on the mechanical properties of prestressed concrete shear walls. The results show that the strain of the joints is less than the yield strain, and the joints do not destroy. The mechanical properties of the joints are similar to those of the cast-in-place reinforced concrete shear walls. Therefore, the performance of the connection could significantly influence the structural performance especially in the final stage in the earthquake.

*1.2. Research Status of the Shear-Type Metal Damper*

Many scholars have conducted extensive research on the behavior of metal dampers used in structural systems. Metal dampers as passive energy dissipation devices commonly serve as non-structural members reciprocating to absorb the input seismic energy and protecting the structural members. This energy dissipation is obtained by plastic deformation in which the structural member is in elastic [19]. Low-yield-point steel has the advantages of low yield strength, large elongation, and good ductility. Its high plastic deformation ability could enhance the structural energy dissipation ability [20]. The shear metal damper proposed by J.M. Kelly was widely used in the damping design of building structures due to simple structure and excellent performance. Whittaker et al. [21] proposed geometrically optimized X-shaped mild steel dampers and triangular soft steel dampers. Zhang and

Zhang [22] experimentally researched different ways of weakening the stiffness of the damper in which the shape in the middle has a great influence on the ductility and the flange plates of different shapes have no obvious influence. Abebe et al. [23–25] conducted experimental research and simulation on the hysteretic behavior of low-yield steel shear dampers. Mortezagholi et al. and Zahrai et al. [26,27] proposed a damper with a circular cross-section by geometrically optimized parameter analysis. In order to solve the connection problem between lead blocks and components, Cheng et al. [28] proposed a baffle-type lead damper and its test results demonstrated excellent energy consumption ability. According to the above study, metal dampers could achieve good energy dissipation ability but their failure would result in degradation of the structural stiffness. To some degree, the high rigidity of in-cast structural systems would limit the performance of dampers and the stiffness degradation of the prefabricated structures would result in the collapse especially in large and super-large earthquakes. U-shaped metal yield damper introduced by Jamkhaneh et al. [29] has been tested about its mechanical displacement, lateral strength, elastic stiffness, and energy dissipation ability. Lin et al. [30] developed a detachable buckling restrained shear plate shock absorber. The influence of key design factors, such as the length-width ratio of the slab and the number of internal composite plates, on the seismic performance of the damper, is studied, and the design formulas for calculating the elastic stiffness and ultimate strength of the damper are proposed. Zhu et al. [31] proposed a metal shear plate damper with an optimized shape. The test results show that the metal shear damper has stable energy dissipation capacity and good low cycle fatigue performance. Belleri et al. [32] proposed that the use of passive energy dissipation and re-centering devices could limit the structural damage. Mazza et al. [33,34] successfully proposed a design procedure for the damper braces to attain a designated performance level according to a certain proportion of reinforcement and further developed a new displacement-based design procedure to proportion hysteretic damped braces considering the effect of a structure's seismic degradation. These procedures are verified to be highly effective when being utilized in designing frame structures.

Some gap dampers are proposed by Rawlinson et al. [35] and De Domencio et al. [36] which could be designed to be engaged after an expected displacement and they depict good performance when being utilized in base-isolated structural system. Therefore, an innovative type of wall-to-wall horizontally connecting structure with high energy dissipation and stiffness lifting ability is proposed, which is expected to enhance the seismic and resilient performance of prefabricated shear wall systems.

This study proposes an innovative "double-step" horizontal-connection and energy-dissipation structure (HES) with the character of high energy dissipation and capacity lifting after the decrease. In its first step, it weakly connects two adjacent shear walls and mainly dissipates the input energy. In its second step, it could strongly integrate two respectively working adjacent shear walls into one unit to obtain one stronger structure member to resist the collapse of the structure system. The design procedure of the HES is briefly depicted in Figure 1. First, a shear walls structural system analytical model is built and analyzed in a predicted earthquake is. After that, the shear bearing capacity $V_f$ and the allowable horizontal displacement $\Delta_{WH}$ of the shear wall are calculated. The shear threshold displacement D of the HES is determined according to the allowable horizontal displacement $\Delta_{WH}$, by which the input energy during the earthquake is dissipated by the ED zone before yield the shear wall. The shear bearing capacity $V_f$ of shear wall is employed to predict the shear capacity of the SL zone of the HES $V_{HSE}$ to ensure that the HES could strongly integrate two respectively working adjacent shear walls into one unit to obtain one stronger structure member to resist the collapse of the structure system. Numerical analysis is performed to comprehensively study the hysteretic behavior of the HES utilizing the validated finite element models. Their hysteretic load-displacement curves, skeleton curves, shear deformation, and failure mode are discussed in detail and the optimized design methods are suggested.

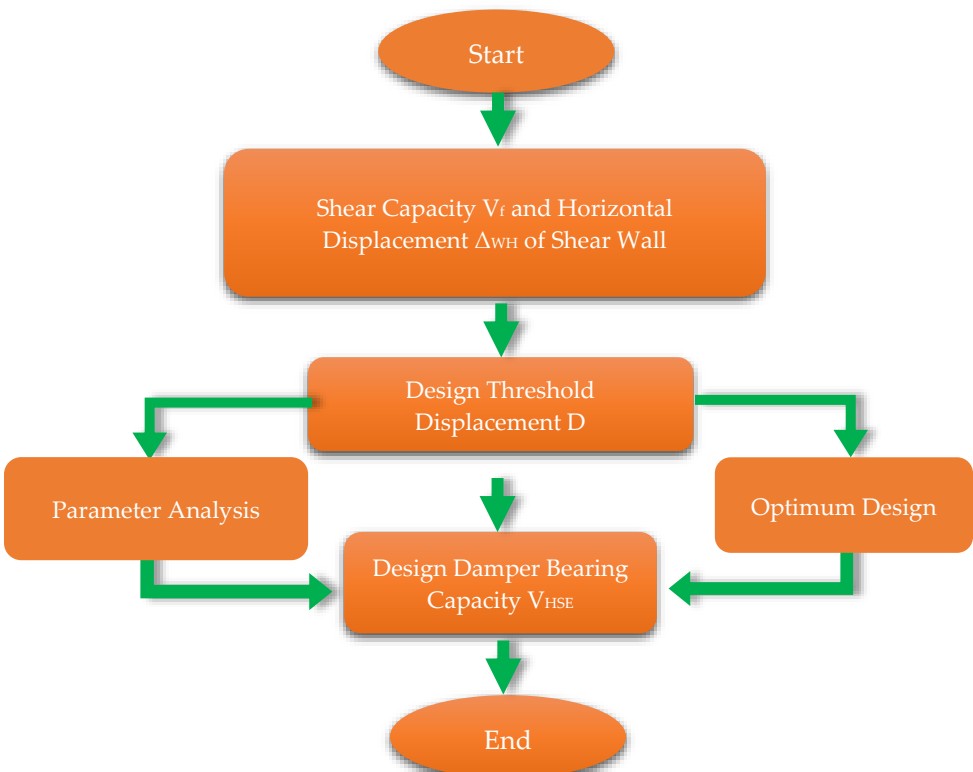

**Figure 1.** Design flow chart of the horizontal-connection and energy-dissipation structure (HES).

## 2. The Mechanism of the HES

### 2.1. General Concepts

The equivalent in-cast connecting method is commonly used in traditional prefabricated shear wall systems which fails to restore the structural function once being damaged in the earthquake. The proposed HES mainly consists of the energy dissipation zone (ED), the stiffness lifting zone (SL) and the horizontal connecting zone (HC) as shown in Figure 2. The HES could horizontally connect two adjacent walls (Figure 2a), the steel plates in the ED zone and SL zone both utilize bolt connection which could be fast replaced after damage in the earthquake and the resilient structural performance is obviously enhanced. The low-yield-strength steel plates are employed in the ED zone mainly dissipating seismic energy. As depicted in Figure 2d–f, the SL zone is composed of the shear stiffness lifting plate, the flange plate, functional bolts, and the buckling restrained plates. The diameter of the functional bolt bar is set to be smaller than that of the circle hole in the shear stiffness lifting plate and this deviated value is the shear displacement threshold. By this threshold control system, the HES exhibits a controlled two-stage mechanical behavior. As depicted in Figure 2, the obvious shear deformation of the HES could be investigated when the in-plane lateral deformation of two adjacent walls are observed in the earthquake. In the first stage, when the shear displacement is smaller than the displacement threshold the functional bolts would not contact the shear stiffness lifting plate and the SL zone has no contribution to the performance of the HES, only the ED zone dissipating the input seismic energy. In the second stage, when the shear displacement is smaller than the displacement threshold, the SL zone would begin to work and the shear stiffness and bearing capacity of the HES increase again. The adjacent walls would be assembled again becoming a strengthening system and the lateral bearing capacity of shear wall system would increase again, which could protect the structure in the large and super-large earthquake.

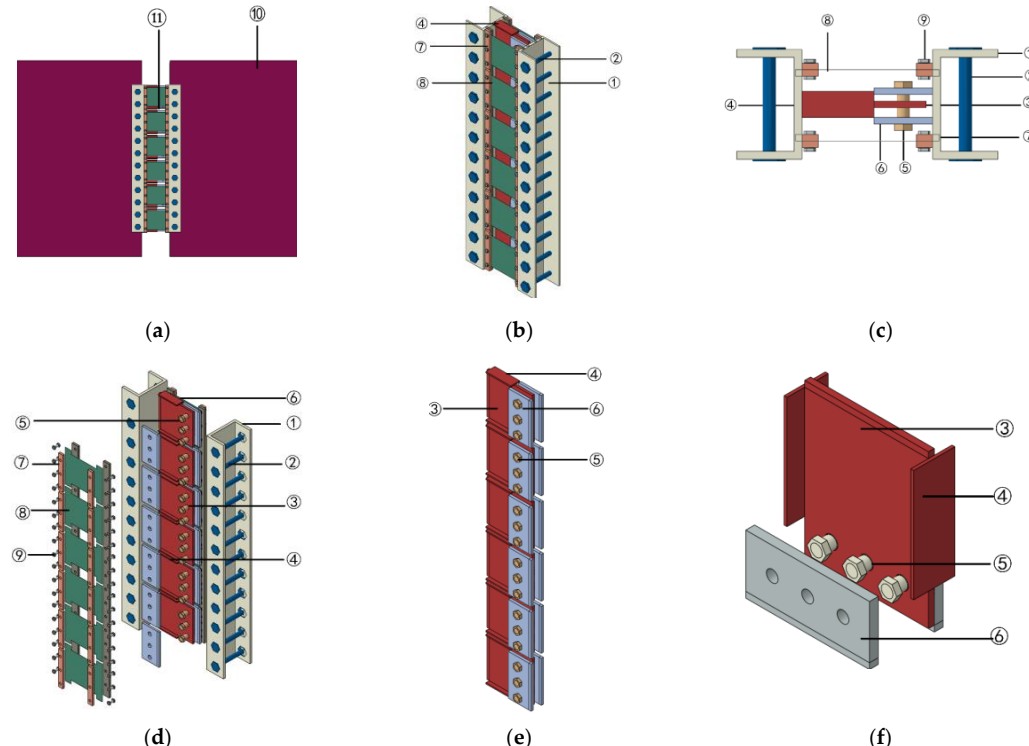

Note: ① Connecting end plate; ② high-strength bolts; ③ the shear stiffness lifting plate; ④ the flange plate; ⑤ functional bolts; ⑥ the buckling restrained plate; ⑦ the backing plateL; ⑧ energy dissipation plate; ⑨ bolts; ⑩ the shear wall; ⑪ the HES

**Figure 2.** Detailed instruction of the HES system. (**a**) The wall connection, (**b**) assembly model, (**c**) cross-section, (**d**) disassembled model, (**e**) lateral stiffness lifting zone, (**f**) threshold control system.

The shear wall and the HES are assumed to be rigid pieces as shown in Figure 3. The deformation of the HES could be computed approximately by Equation (1) when achieving the horizontal displacement of shear walls.

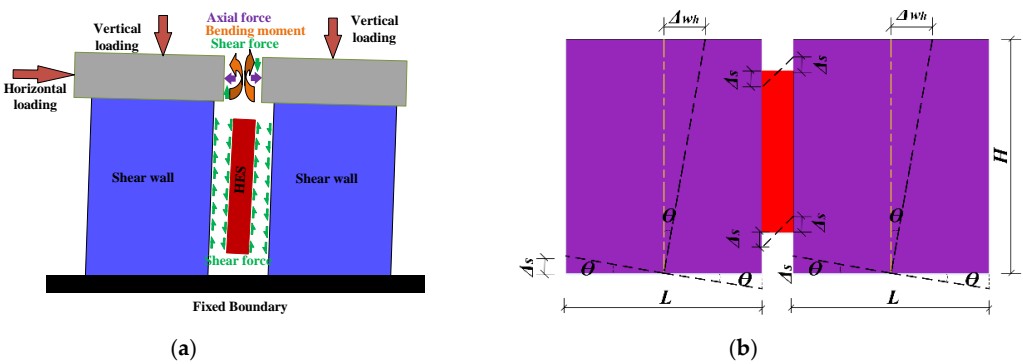

**Figure 3.** The shear deformation of the HES. (**a**) The boundary condition, (**b**) calculation shear deformation of the HES according to the drift displacement.

The drift ratio is a significant parameter defined in many specifications, which is utilized to evaluate structural performance. Therefore, the limited shear deformation of the HESs could be calculated.

$$\Delta_s = 0.5 \cdot \theta \cdot L = 0.5 \cdot \Delta_{hw} \cdot L / H \tag{1}$$

where, $\Delta_{hw}$ is the horizontal displacement of the shear wall, $\Delta_s$ is the shear deformation of the shear wall, $\theta$ is the drift ratio, L is the width of the shear wall, H is the height of the shear wall.

## 2.2. The Expected Failure Mode of the HES

The low-yield-point steel plates (LY 100) the yield stress of which is 100 MPa, are utilized in the ED zone. In order to ensure the bearing capacity and shear stiffness of the SL zone, Q345B is utilized the yield stress of which is 345 MPa. The threshold control displacement should be reasonably established to ensure that the ED zone and the SL zone could perform well in sequence. Furthermore, the threshold displacement could be adjusted according to the requirement of the expected structural performance, by changing the hole diameter in the shear stiffness lifting plate. In addition, the shear stiffness and strength of the shear stiffness lifting plate could be effectively enhanced by adding the flange plate on both sides. Simultaneously, the functional bolts are designed to come to failure in the second stage and the other parts of the SL zone are in elastic. When the ED zone and the functional bolts are damaged in the earthquake, they could be fast and easily replaced, largely enhancing the structural resilient performance. Therefore, the ED zone is expected to dissipate the input seismic energy firstly and the functional bolts are expected to fail with high ductility.

In order to clarify the mechanical mechanism of the HES, simplified models are used in finite element modelling analysis. The lateral connection of the HES to the wall is considered to be rigid. The bolted connection of the steel plates in the ED zone is also considered to be a rigid connection. The threshold control displacement of the typical specimen of HES is set to be 3 mm. On the basis of the simplification, this expected failure mode would be validated by numerical analysis.

## 3. Model Development and Validation

The finite element analysis is an efficient way to predict the performance of testing specimens. The software ABAQUS was employed to simulate the typical hysteretic behavior of the HESs. This numerically modelling method is validated by successfully simulating the cyclic behavior of one low yield strength steel shear plate damper tested by Zhang et al. Using this modelling method, the performance of the HESs are predicted and evaluated.

### 3.1. Finite Element Model for The HESs

The model and the boundary conditions are depicted in Figure 4. The shell element S4R is used to model the behavior of the steel plates in the ED zone including their buckling. Eight-node-three-dimensional solid element (C3D8R) with reduced integration and hourglass control is utilized to simulate functional bolts, the flange plate and the shear stiffness lifting steel plates. The overall meshed model includes a total of 40,444 elements. A typical surface-to-surface contact with a penalty algorithm is employed between the functional bolts and the shear stiffness lifting plate. A hard contact pressure-over closure relationship is adopted to model the normal contact behavior and the friction coefficient is set to be 0.2 to simulate the tangential slip behavior. The same contact setting is used between the surfaces of buckling restrained steel plates and shear stiffness lifting steel plates. In addition, the geometric nonlinearity is considered to model the intermittent contact behavior in the SL zone. The circular part and the bolts in the SL zone are more elaborately meshed. Considering the computational efficiency and reliability the mesh size of the HES is the same as above which is appropriate for this analysis.

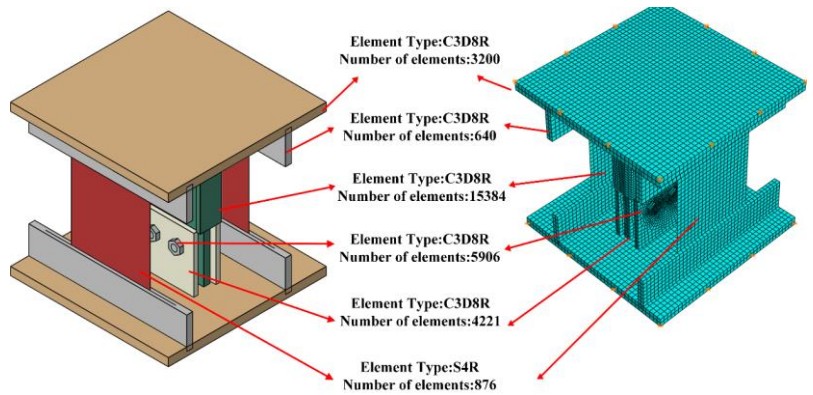

**Figure 4.** The finite element model for the HES.

### 3.2. Material Constitutive Models

The von Mises yield criterion and bilinear model are utilized for simulating the behavior of the ED zone made of low yield point steel (LP100) and the SL zone made of Q345B. When the strain $\varepsilon$ of the steel sheet is less than the ultimate strain, the actual stress $\sigma$ of the steel sheet is equal to the elastic modulus multiplied by the strain $\varepsilon$; when the strain $\varepsilon$ of the steel sheet is greater than the ultimate strain, the actual stress $\sigma$ of the steel sheet is equal to the elastic modulus multiplied by the ultimate strain. With the advancement of structural health monitoring technology, high precision strain measurement can be obtained [37,38] to guarantee the quality and reliability of stress calculation under complicated load bearing situation. The elastic modulus of the steel is set to be = $2.05 \times 10^5$ MPa, the Poisson's ratio of steel is taken as 0.3 [39,40].

### 3.3. Validation of the Finite Element Model

#### 3.3.1. Verification

Zhang et al. [41] conducted a cyclic fatigue performance test of a low yield strength steel shear plate damper and the failure mode is depicted in Figure 5b investigating obvious buckling. Using the method above, the finite elemental analysis (Figure 5a) is conducted and the results including the failure mode and the cyclic curve are shown in Figure 5c,d. It can be seen that simulated buckling deformation is consistent with the tested. The simulated curve agrees well with the test curve but a little deviation of the initial stiffness could be observed which might be caused by failing to accurately model the actual loading boundaries. In addition, this finite element model(FEM) accurately predicts the bearing capacity in each loading circle. Therefore, this modelling method could be used to predict the behavior of the HESs and the results could be used to evaluate their performance.

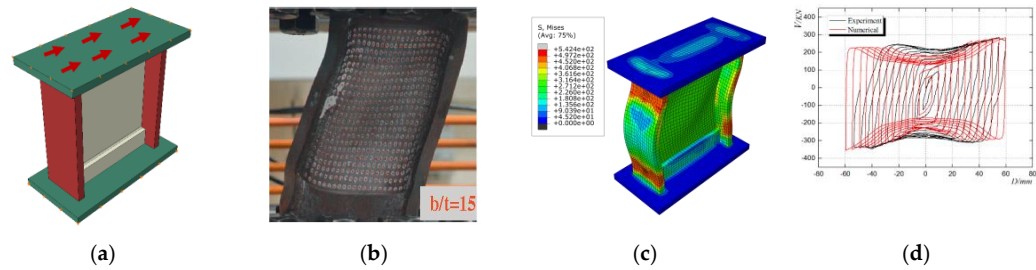

| (a) | (b) | (c) | (d) |

**Figure 5.** The validation of the finite element modeling method. (**a**) Finite element model, (**b**) tested failure mode, (**c**) simulated failure mode, (**d**) hysteresis curves.

Xu et al. [39] proposed a metal shear plate damper utilizing the low yield performance of BLY 160 materials for effective energy dissipation and conducted hysteric tests to evaluate its performance.

This metal shear plate damper is composed of four components: shear panels, confined flanges, stiffening ribs and the roof/floor plates for connection (Figure 6b). To validate the simulating method used above, the above modelling method is employed to carry out a finite element analysis of the damper (Figure 6a). The failure mode, the buckling behavior and numerical simulated curves are shown in Figure 6c,d. It is found that the simulated curve has good coincidence with the test curve, but the load-displacement amplitude has some deviation. Some unpredicted slip occurred in the boundary during the experimental loading process, resulting in a deviation between the finite element loading displacement and the test. Therefore, the modelling method could simulate the hysteric behavior of the HESs and could be utilized to evaluate their performance.

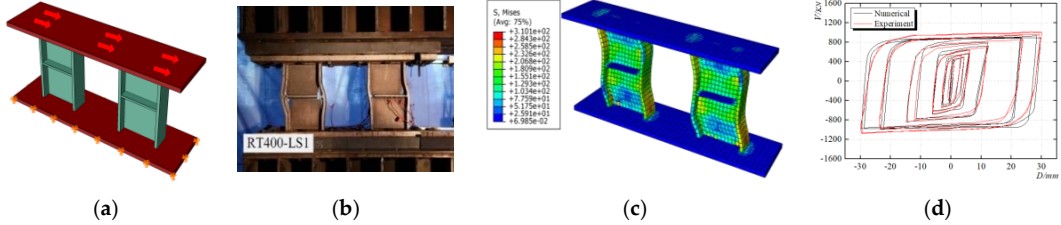

(**a**)            (**b**)            (**c**)            (**d**)

**Figure 6.** The validation of the finite element modelling method. (**a**) Finite element model, (**b**) tested failure mode, (**c**) simulated failure mode, (**d**) hysteresis curves.

### 3.3.2. The Simulation of the Double-Step Performance of the HESs

The detailed geometric parameters are listed in Table 1, Table 2 and Figure 7. The monotonic behavior of specimen HES1 is simulated and the results are shown in Figure 7 and its displacement threshold is set as 3mm. When the shear displacement is applied to 0.24 mm (the drift ratio = 0.12/100), the steel plate in the ED zone begins to yield and the yielding area gradually enlarges from the two ends of the steel plate to the middle part as shown in Figure 8a. With the increment to 3 mm (the drift ratio $\theta$ = 1.5/100), the steel plate totally comes into plastic and the shear strength of HES1 comes to the end of its first step, as shown in Figures 8b and 9a. With the development of deformation, the SL zone begin to work and when the horizontal displacement is up to 3.8 mm (the drift ratio = 1.9/100), the middle part of functional bolts yields, the shear stiffness lifting plate is almost in elastic with slight stress concentration at the bolt holes as shown in Figure 8c,d. Finally, the full section of the functional bolts come into plastic and the shear stiffness lifting plate locally yields, as shown in Figure 8e,f. The top and bottom boundaries utilized thick steel plates to simulate the connection to shear walls. When the HES is employed in steel composite shear wall system [42] and steel-damping-concrete composite wall sytems [43] the high strength bolt connection is available.

**Table 1.** Common geometric parameters for specimens.

| The Typical Dimensions for All HES Specimens (mm) | | | | | | | | | | | | |
|---|---|---|---|---|---|---|---|---|---|---|---|---|
| b | H | t | a | h | $h_c$ | $t_c$ | $b_a$ | $h_a$ | $t_a$ | $b_f$ | $h_f$ | $t_f$ |
| The HESs | 330 | 380 | 400 | 65 | 20 | 70 | 10 | 40 | 200 | 56 | 200 | 140 | 10 |

**Table 2.** Specific dimensions for HES specimens.

| No. | The Geometric Parameters of HES Specimens (mm) | | | | | | | | | | | | | |
|---|---|---|---|---|---|---|---|---|---|---|---|---|---|---|
| | $b_w$ | $h_w$ | $t_w$ | $b_t$ | $h_t$ | $t_t$ | $R_1$ | $R_2$ | $D_{(R1 - R2)}$ | $R_3$ | $b_{w1}$ | $h_{w1}$ | $b_{w2}$ | $h_{w2}$ |
| HES1 | 200 | 200 | 10 | 200 | 330 | 16 | 11 | 8 | 3 | - | - | - | - | - |
| HES2 | 200 | 200 | 10 | 200 | 330 | 16 | 11 | 8 | 3 | - | 50 | 100 | - | - |
| HES3 | 200 | 200 | 12 | 200 | 330 | 16 | 11 | 8 | 3 | - | 50 | 100 | - | - |
| HES4 | 200 | 200 | 10 | 200 | 330 | 16 | 10 | 8 | 2 | - | - | - | - | - |
| HES5 | 200 | 200 | 10 | 200 | 330 | 16 | 13 | 8 | 5 | - | - | - | - | - |
| HES6 | 200 | 200 | 10 | 200 | 330 | 16 | 11 | 8 | 3 | - | - | - | 50 | 50 |
| HES7 | 200 | 200 | 10 | 200 | 330 | 16 | 11 | 8 | 3 | 50 | - | - | - | - |

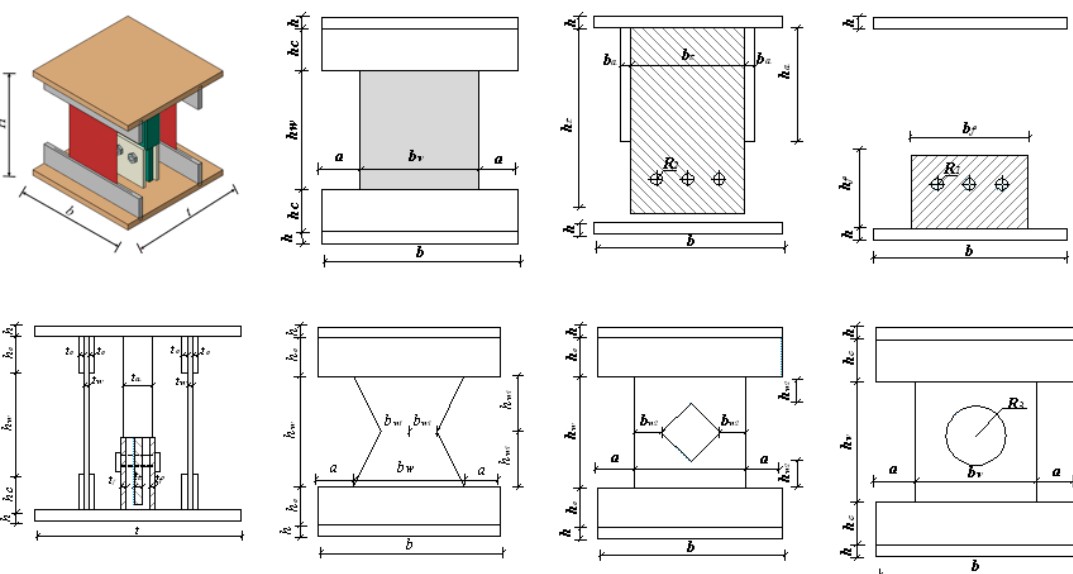

**Figure 7.** The detailed dimensions of HES specimens. (**a**) 3D model, (**b**) energy dissipation (ED) zone/rectangular steel plates (HES1/HES4/HES5), (**c**) stiffness lifting (SL) zone/the shear stiffness lifting plate, (**d**) SL zone/the shear stiffness lifting plate, (**e**) the top view, (**f**) ED zone/X-shaped steel plates(HES2/HES3), (**g**) ED zone/Rectangular hole (HES6), (**h**) ED zone/Circular hole (HES7).

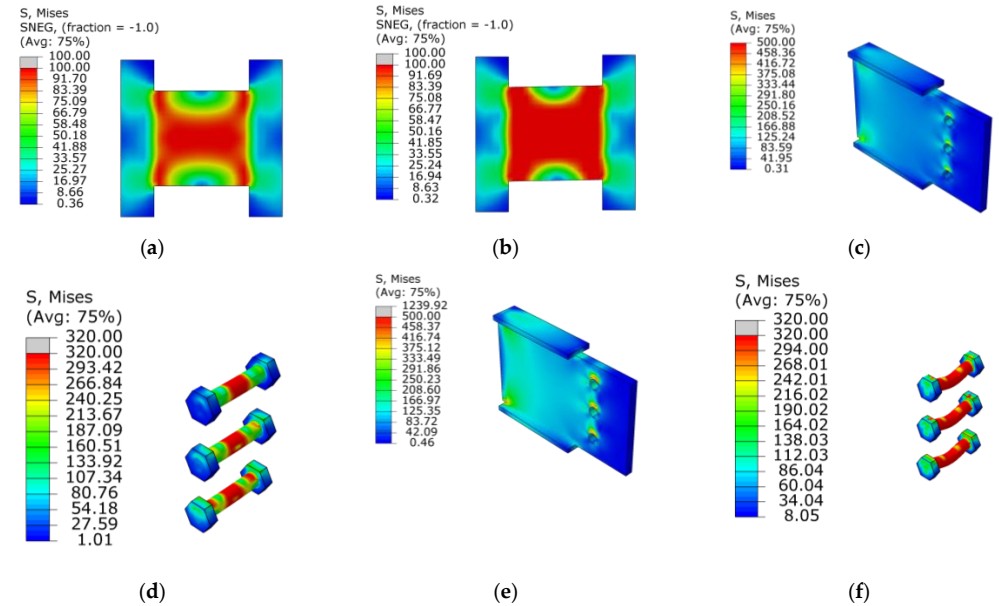

**Figure 8.** The validation of the finite element modelling method. (**a**) The stress distribution of ED zone ($\theta = 0.12\%$), (**b**) the stress distribution of ED zone ($\theta = 1.5\%$), (**c**) the stress distribution of SL zone ($\theta = 1.9\%$), (**d**) the stress distribution of functional bolts ($\theta = 1.9\%$), (**e**) the stress distribution of SL zone($\theta = 5\%$), (**f**) the stress distribution of functional bolts ($\theta = 5\%$).

The double-step force-displacement curve of HES1 is depicted in Figure 9a and the typical double-step working mechanism of the HESs is shown in Figure 9b. The performance of the HES can be observed with four stages including the completely elastic stage, ED plastic stage, SL elastic stage and functional bolts plastic stage. In the completely elastic stage (the OA line in Figure 9a), steel plates in ED are in elastic and the initial shear stiffness is K1 = 525.19 kN/mm. When coming to the ED plastic stage (the AB line), the input seismic energy is mainly dissipated by the plastic deformation. From point B to point C (the SL elastic stage), the SL zone is almost in elastic and the HES restores the bearing

capacity and the shear stiffness with K2 = 166.32 kN/mm. As the shear displacement increases, the bending deformation of the functional bolts gradually develops into the fourth stage (the CD line) and finally the come into yielding as depicted in Figure 9b.

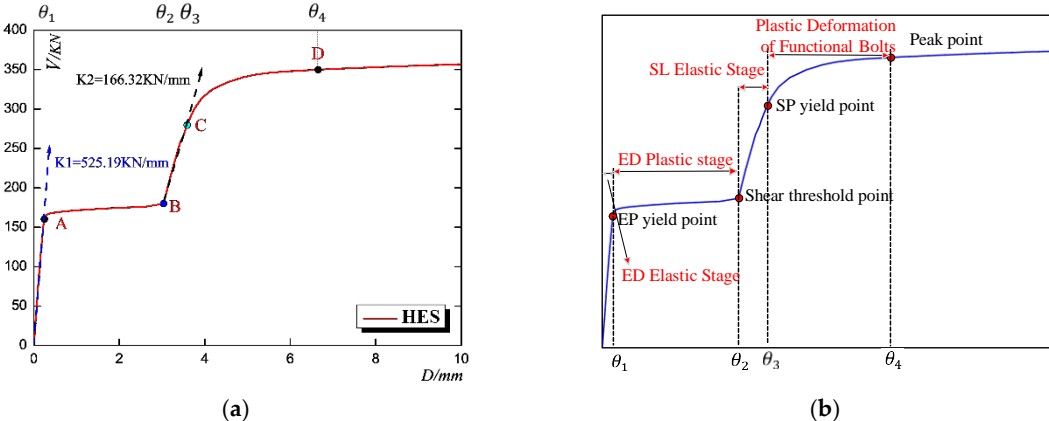

(a)  (b)

**Figure 9.** The typical curve for the HES. (**a**) Monotonic force-displacement curve, (**b**) the failure mode.

### 3.3.3. The Typical Hysteric Behavior of the HESs

The loading protocol is shown in Figure 10a which is divided into ten stages with two cycles in each stage. In this protocol, two types of calculating the loading displacement are adopted respectively considering the character of the ED zone and the SL zone. In the stages of only the ED zone working, the loading displacement is set according to the yield displacement of the steel plate and the latter loading amplitude is twice the amplitude of the previous loading displacement. In the stages of the ED zone and the SL zone working simultaneously, the loading displacement is set according to the yield displacement of the SL zone and the latter loading amplitude is 1.4 times of the previous loading amplitude. The simulated typical hysteresis curve is depicted in Figure 10b demonstrating the deforming characteristics and energy dissipation performance. Due to the in-plane shear resistance of the steel plate in the ED zone, the HES exhibits a character of the large initial stiffness and the high energy dissipation ability. It can be seen that the cyclic curves are close to rectangular shapes indicating the great energy dissipation ability. When the loading displacement is larger than the designed displacement threshold, the area of the hysteresis curve and the bearing capacity both gradually increase largely. The hysteretic curve exhibits double-step operating characters with both high energy dissipation ability and the shear stiffness re-lifting ability.

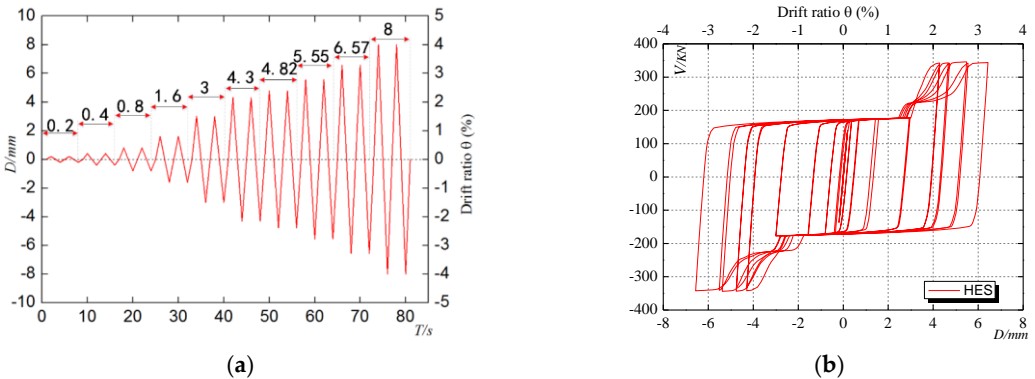

(a)  (b)

**Figure 10.** The typical hysteresis curve of the HESs. (**a**) The loading protocol, (**b**) the hysteretic curve.

## 4. Parametric Analysis

In order to further optimize the performance of the HES, seven specimens with different threshold displacements, different steel plate width-thickness ratios and different shapes of the steel plates in the ED zone are designed and simulated using software ABAQUS. The detailed geometric parameters are shown in Figure 7, Table 1, and Table 2. The common and specific geometric parameters are respectively listed in Tables 1 and 2. The influence of the shape of the steel plate in the ED zone is investigated by the comparison among specimens HES1, HES2, HES6, and HES7. The influence of the width-thickness ratio is studied by comparing the performance of HES2 and HES3. The threshold displacements of HES2, HES4, and HES5 are changed to investigate their influence.

### 4.1. Parameter Analysis Under Monotonic Load

### 4.1.1. The Investigation on the Influence of the Shape of the Steel Plate in the ED Zone

The monotonic behavior of four specimens with different shape types for steel plated in the ED zone including the rectangular shape (HES1), the X-type (HES2), the rectangular shape with one diamond-shaped hole (HES6) and the rectangular shape with one circular hole (HES7) are simulated. Their load-displacement curves, the stress distribution of the steel plates in the ED zone and typical analyzed results are depicted in Figure 11 and Table 3.

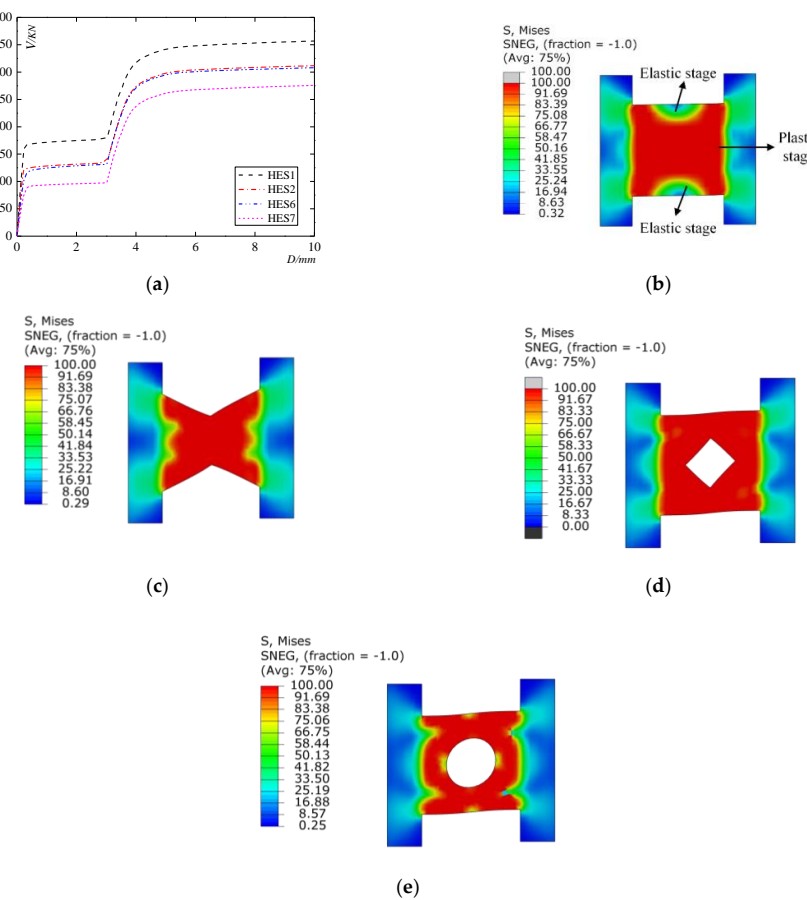

**Figure 11.** The influence of the shape of the steel plate in the ED zone. (**a**) Load-displacement curves, (**b**) the steel plates in ED zone (HES1), (**c**) the steel plates in ED zone (HES2), (**d**) the steel plates in ED zone (HES6), (**e**) the steel plates in ED zone (HES7).

**Table 3.** Typical analyzed results.

| NO. | The Surface Area (mm²) | The Initial Stiffness (KN/mm) | Yield Load (KN) | Yield Displacement (mm) |
|---|---|---|---|---|
| HES1 | 4.00 | 525.19 | 160.05 | 0.304 |
| HES2 | 3.00 | 458.13 | 116.19 | 0.253 |
| HES6 | 3.50 | 370.01 | 95.78 | 0.258 |
| HES7 | 3.21 | 319.78 | 70.63 | 0.221 |

It is observed in Figure 11a that the bearing capacity the HES with rectangular steel plates in the ED zone is the largest. Among the four shape-types of the steel plates in the ED zone, the bearing capacity of the rectangular typed specimen is larger than those others but this type of shape failed to fully develop the plastic deformation. Among the three optimized shapes, the X-shaped specimen HES2 with the smallest surface area exhibits the best performance including the initial stiffness and bearing capacity as shown in Table 3.

### 4.1.2. The Influence of the Width-Thickness Ratio of the Steel Plate in the ED Zone

The monotonic behavior of specimens with optimized X-shaped low-yield-point steel plates (HES2 and HES3) are analyzed and the load-displacement curves are shown in Figure 12a. The thickness of X-shaped low-yield steel plates of specimen HES2 and HES3 are respectively set to be 10 mm and 12 mm which is the only difference between them. With the increase of thickness, the obvious increase of shear capacity (22.5%) can be investigated and the initial shear stiffness is slightly increased.

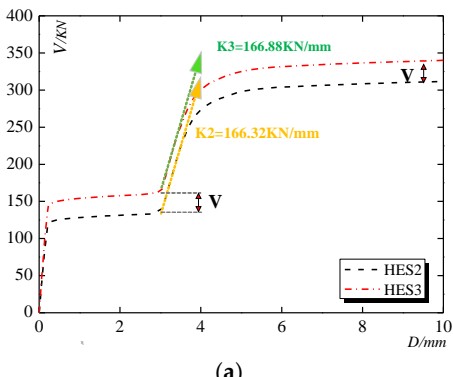
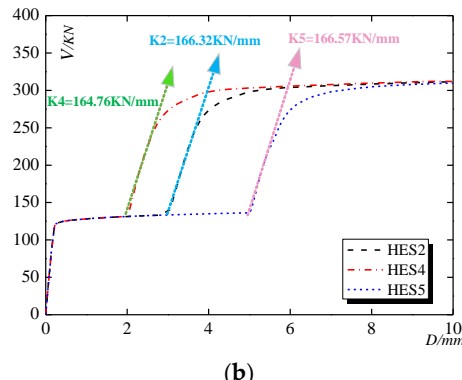

(**a**)        (**b**)

**Figure 12.** The monotonic load-displacement curves for HES2/HES3/HES4/HES5. (**a**) The influence of the width-thickness ratio, (**b**) the influences of the displacement of threshold.

### 4.1.3. The Shear Stiffness Lifting Control System

The shear displacement threshold is a significant parameter to decide on which level of shear deformation the SL zone begins to bear loads. The shear displacement threshold ($D_{(R1-R2)}$) of specimen HES2, HES4 and HES5 are respectively set as 3 mm, 2 mm and 5 mm and the computed load-displacement curves are shown in Figure 12b. It is concluded that this shear stiffness control system endows the HES with double-step character, sufficient energy dissipation ability and the ability to prevent the collapse of the structure in large earthquakes. If the shear displacement threshold is too large, the strength degradation of the steel plate in the ED zone will result in a decrease in the ultimate strength of the HES. Therefore, the shear displacement threshold could be adjusted according to the requirement of performance design.

### 4.2. Simulated Hysteretic Curves

As depicted in Figure 13, the shapes for the simulated hysteretic curves of the seven specimens are similar to each other and. Due to the shear stiffness lifting control system, two stages of energy

dissipation and load-bearing is investigated. When shear displacement is smaller than the threshold displacement, the shape of the hysteretic curves is similar to rectangular and the high energy dissipation performance of the ED zone is obtained. With the increase of the thickness of steel plates, the bearing capacity and the energy dissipation ability are both enhanced. The Bouc–Wen–Baber–Noori model could be adopted to describe the hysteric character and used to analyze the seismic behavior of wall systems. The gap in the SL zone could be further utilized to add some viscous damping material to increase the damping ratio according to the requirement of the wall system.

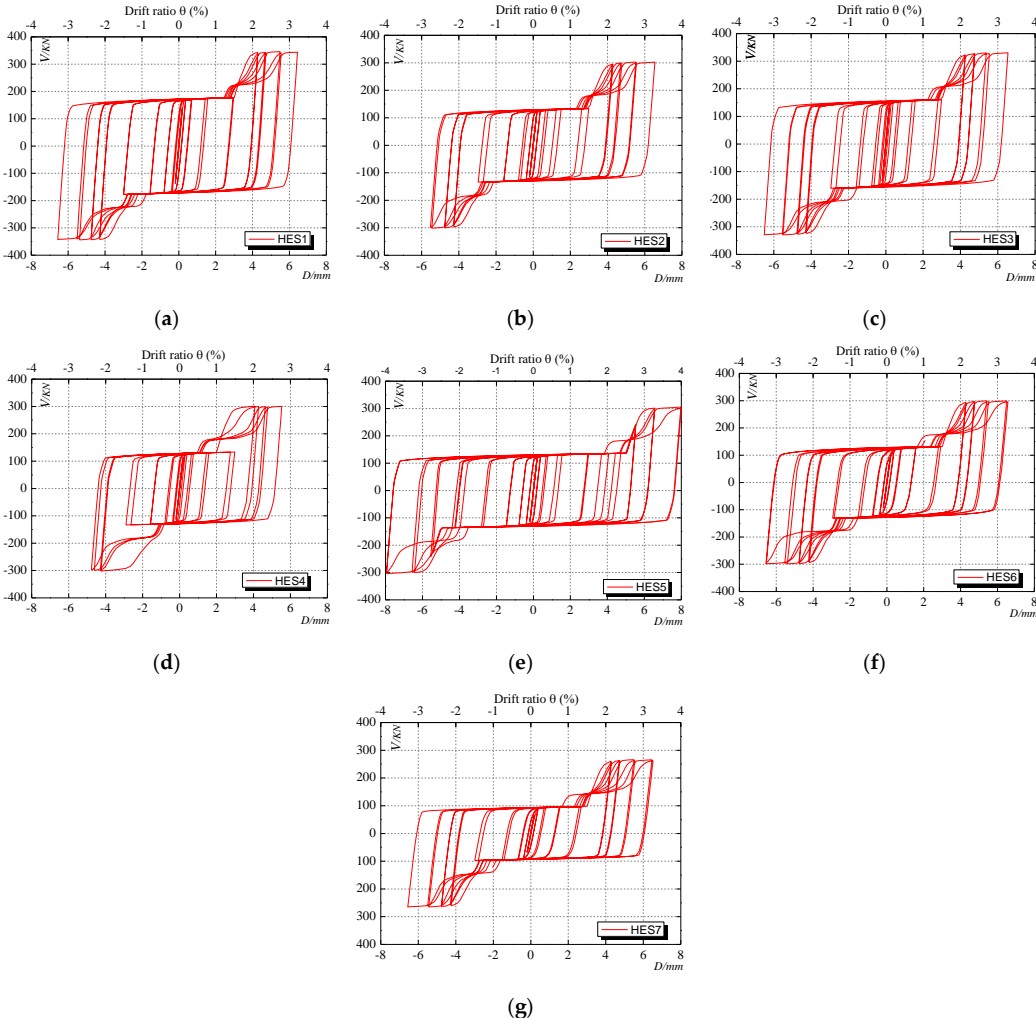

**Figure 13.** The hysteretic curves of specimens. (**a**) HES1, (**b**) HES2, (**c**) HES3, (**d**) HES4, (**e**) HES5, (**f**) HES6, (**g**) HES7.

*4.3. Skeleton Curves*

The skeleton curves (Figure 14a) are obtained by the peak point of the envelope in the first cycle of each loading step which could be used to evaluate the performance of the strength, shear stiffness and ductility. The double-step mode of the seven specimens is basically coincident which is mainly controlled by the shear displacement threshold. The comparison HES structural specimens with different threshold displacements, different thicknesses, and shapes of steel plates in the ED zone are respectively shown in Figure 14b–d. It can be seen that the energy-dissipation process of the ED zone is slightly extended as the threshold displacement increases. However, the bearing capacity of the SL zone is not obviously influenced by the threshold displacement. With the increase of thickness, both the bearing capacity of the ED zone and the HES increases. The yield displacement depends on neither

the thickness nor the bearing capacity. The X-shaped steel plates with the smallest surface area exhibit the highest bearing capacity in shapes optimized from the rectangular shape which is suggested to be utilized in the ED zone.

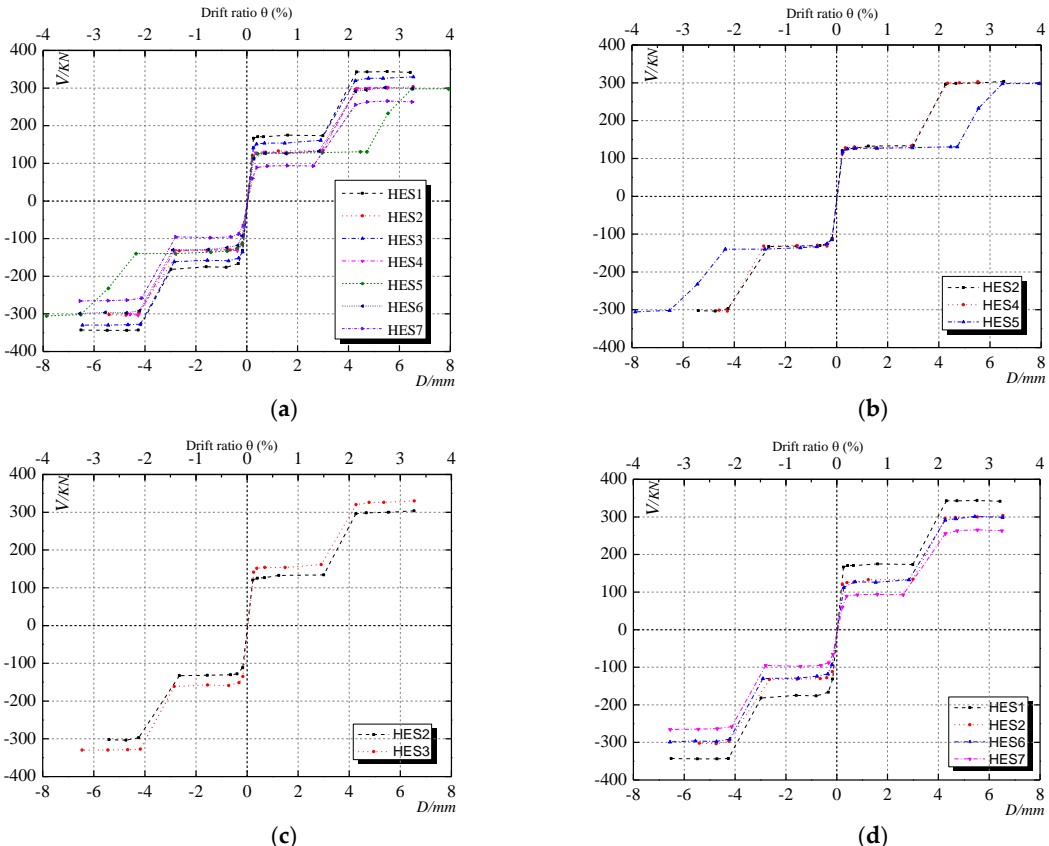

**Figure 14.** Skeleton curves. (**a**) HES1-HES7, (**b**) HES2/HES4/HES5, (**c**) HES2/HES3, (**d**) HES1/HES2/HES6/HES7.

### 4.4. The Energy Dissipation Ability of the ED Zone

It can be seen from Figure 15a that, basically, the HES specimens with the rectangular shape of steel plates in the ED zone exhibit larger bearing capacity in the same deformation. But the rectangular steel plate would result in stress concentration and the failure in the bolt connection boundary. The specimen HES2 with optimized X-shape steel plates in the ED zone exhibits the highest energy dissipation capacity compared with that of specimen HES6 and HES7. Because of the thickness increase of steel plates in the ED zone, the energy dissipation capacity of specimen HES3 is larger than that of specimen HES2 as shown in Figure 15b. Figure 15c depicts that the specimens with the larger threshold displacement would dissipate less energy before the SL zone coming to work. When the SL zone begins to bear the load and the plastic deformation of the bolts would increase its energy dissipation ability.

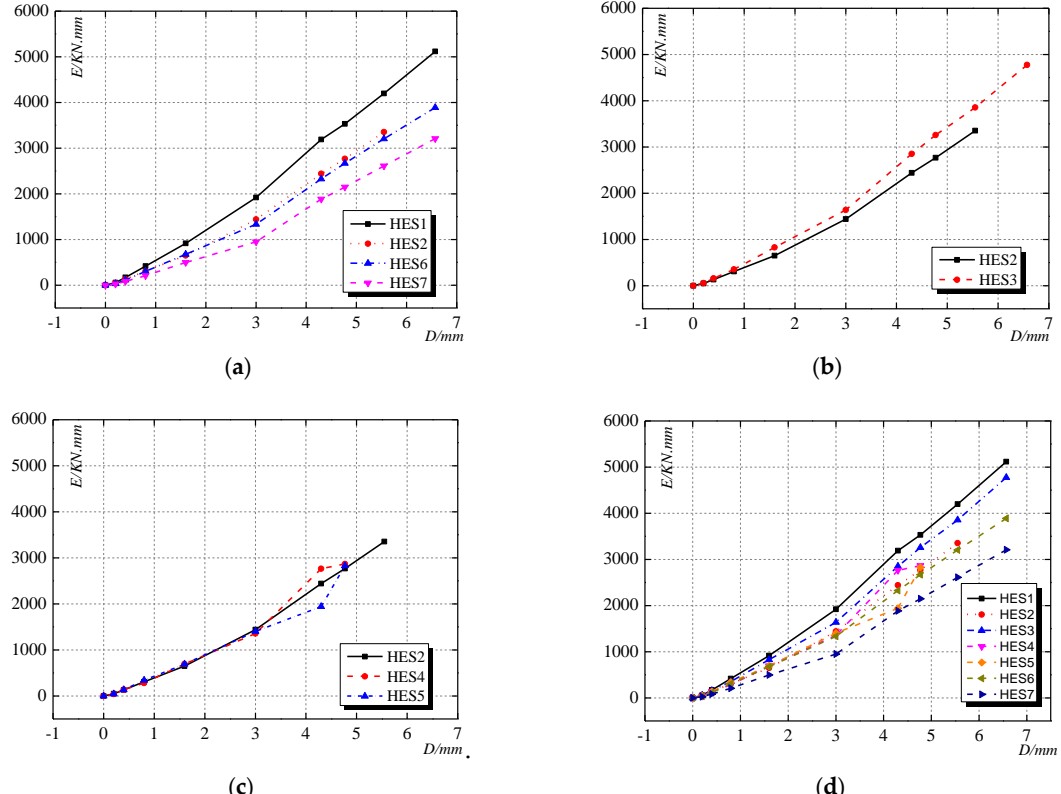

**Figure 15.** Accumulative energy dissipation curves. (**a**) HES1/HES2/HES6/HES7, (**b**) HES2/HES4/HES5, (**c**) HES2/HES3/HES4, (**d**) HES1-HES7.

## 5. Summary and Conclusions

The inherent out-of-plane stiffness and strength of the HES are mainly provided by the steel plates in the ED zone. The strong connections to the two adjacent walls of the HES can also ensure its in plane performance. In the further study a kind of supporting structure will be developed in the SL zone to enhance its out of place performance. This study mainly analyzed the in-plane monotonic and hysteretic behavior of the HESs using software ABAQUS. Seven specimens of the HES are designed with different parameters and the influence of the parameters on their performance is investigated giving some optimized suggestions. The failure mode of the HESs is observed and their typical performance load-displacement is proposed with the character of double-step. Because of the design of the shear stiffness lifting control system, the ED zone would firstly come into plastic dissipating the input seismic energy and the SL zone would come into play when the large deformation occurs in a large and super large earthquake. Therefore, the HES can be used as the horizontal connecting member for the shear wall system and simultaneously enhance its seismic and resilient performance. On the basis of the above simulation and analysis, the following conclusions are obtained.

(1) The proposed shear displacement threshold control system endows the HES with the ability of energy dissipation, stiffness lifting and shear strength lifting by the separate function of the ED zone and the SL zone. The bolt connection in the ED zone and the functional bolts could be easily and rapidly replaced when being damaged in the earthquake, which largely enhances the resilient performance and the recovery capability of the structural system. The threshold can be adjusted according to the requirement of the structural performance, this proposed The HES could be used in prefabricated shear wall system and the performance-based design could be applied.

(2) The rectangular shape for the steel plate in the ED zone exhibits good energy dissipation performance and is easy for construction. According to the parameter analysis of the shape, the

X-shaped steel plate in the ED zone exhibits the best performance and this type is suggested to be utilized in the HESs.

(3) The shear deformation of the HES is caused by the horizontal displacement of the shear walls. When being employed in shear wall structures, the ultimate drift ratio of the HES in this study is about 4%, which could be adjusted to meet the requirement of corresponding horizontal displacement of the shear wall. The ductility coefficient of the steel plate in the energy dissipation zone is about 15 and the use of the low-yield-point steel could effectively enhance the energy dissipation ability in small shear deformation during small earthquakes.

**Author Contributions:** C.Z. and L.Z. provided the conceptual design of the novel composite wall; L.Z. and L.K. conceived and designed the FE models; L.Z. and L.K. analyzed the data; L.Z. and L.K. wrote the paper; L.Z. and C.Z. revised the paper. All authors have read and agreed to the published version of the manuscript.

**Funding:** This research was funded by National Natural Science Foundation of China (Grant No. 51708318, 51678322), the International Postdoctoral Exchange Fellowship Program (Grant No. 20190015), and the Ministry of Science and Technology of China (Grant No. 2017YFC0703603).

**Acknowledgments:** The research is financially and technically supported by the Taishan Scholar Priority Discipline Talent Group program funded by the Shandong Province, the Cooperative Innovation Center of Engineering Construction and Safety in Shandong Blue Economic Zone scheme funded by the Shandong Province and the first-class discipline project funded by the Education Department of Shandong Province.

**Conflicts of Interest:** The authors declare no conflict of interest.

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
