# Peer review of "Numerical Study on Hysteretic Behaviour of Horizontal-Connection and Energy-Dissipation Structures Developed for Prefabricated Shear Walls"

_applsci, doi:10.3390/app10041240_

Round 1

Reviewer 1 Report

The main finding of the study presented in the paper consists in the proposal of a new double-step dissipative device named HES (i.e. Horizontal-connection and Energy-dissipation Structure), for the seismic protection of prefabricated reinforced concrete shear walls. To this end, finite element modelling of the HES is proposed and validated through experimental results available in literature. Then, a parametric investigation is carried out in order to investigate main parameters: e.g. shape and width-thickness ratio of the steel plate in the energy dissipation zone; shear stiffness lifting control system. The paper is well supported from a theoretical point of view, and exhaustive in explaining the objects of the study and its results. The topic is timely and of interest for readers. Abstract and conclusions are adequate. The quality of Figures is good. Based on these observations, in my opinion the manuscript could be accepted for publication, provided that revisions are introduced in it, according to the minor suggestion listed below.

1.Literature review is good. However, some recent works presenting displacement-based design procedures of dissipative devices should be added, with a view to the proposal of a design procedure of the HES. The following recent papers are suggested to complete references:

Mazza, F. A simplified retrofitting method based on seismic damage of a SDOF system equivalent to a damped braced building. Engineering Structures, 2019, 200, 109712, doi:10.1016/j.engstruct.2019.109712.

Mazza, F.; Vulcano, A. Displacement-based design procedure of damped braces for the seismic retrofitting of r.c. framed buildings. Bulletin of Earthquake Engineering, 2015, 13, 2121-2143. doi: 10.1007/s10518-014-589 9709-7.

Author Response

We appreciate referee 1’s positive comments, especially regarding the importance of our work, and giving encouragements, such as “The paper is well supported from a theoretical point of view, and exhaustive in explaining the objects of the study and its results.”.

Reviewer 2 Report

The paper is well-written and original. More effort is required about verification/validation of this study. Only Figure 4d depicts a pertinent comparison of the proposed model with literature where even in this case, the divergence for results is obvious. Please provide one more example in order to provide more acceptability criteria for the readers.

Author Response

We thank the referee for the positive comments and review. Those comments are very valuable and helpful for revising, improving our paper and the further research, as well as the important guiding significance to our researches.

Reviewer 3 Report

The paper entitled "Numerical Study on Hysteretic Behaviour of Horizontal-connection and Energy-dissipation Structures Developed for Prefabricated Shear Walls" is a good and interesting research. It is suitable for publication.

Author Response

We thank the referee for the comments and review. Those comments are very valuable and helpful for revising, improving our paper and the further research, as well as the important guiding significance to our researches. Our responses to the referee 3’s suggestions are as follows:

Comments: The paper entitled "Numerical Study on Hysteretic Behaviour of Horizontal-connection and Energydissipation Structures Developed for Prefabricated Shear Walls" is a good and interesting research that proposes a developed horizontal-connection and energy-dissipation structure (HES), which could be employed for horizontal connection of the prefabricated shear wall system. It is suitable for publication after some corrections.

Point 1:

Page 2, Line 56: “Soudki et al.”; it is better “Noel and Soudki”;

Response 1:

The revised portion are as follows: Noel and Soudki performed a reciprocating loading test on prefabricated shear walls and found that the bearing process of the horizontal joint could be defined as three stages which are the elastic stage before slipping, the elastoplastic stage before the damage of horizontal joint and the total slip damage. (in Section Introduction)

Point 2:

Page 2, Line 62: “Kurama et al.”; it is better “Smith and Kurama”;

Response 2:

The revised portion are as follows: Smith and Kurama studied the prestressed specimens and found that their strength and initial stiffness are similar to those of cast-in-place specimens. The test piece demonstrated slight damage with a large nonlinear displacement, good self-centering ability but a little decrease in energy dissipation ability.

Point 3:

Page 2, Line 90: “Whittaker”; it is better “Whittaker et al.”;

Response 3:

The revised portion are as follows: Whittaker et al [21] proposed geometrically optimized X-shaped mild steel dampers and triangular soft steel dampers.

Point 4:

Page 3, Line 95: “Mortezagholi et al.”; it is better “Mortezagholi et al. and Zahrai et al.”;

Response 4:

The revised portion are as follows: Mortezagholi et al. and Zahrai et al. proposed a damper with a circular cross-section by geometrically optimized parameter analysis.

Point 5:

Page 3, Line 32: “Belleri”; it is better “Belleri et al.”;

Response 5:

The revised portion are as follows: Belleri et al. [32] proposed that the use of passive energy dissipation and re-centring devices could limit the structural damage.

Point 6:

Page 4, Line 160: “Figure 14”; it is probably “Figure 2”;

Response 6:

The revised portion are as follows: The shear wall and the HES are assumed to be rigid pieces as shown in Figure3(One more figure was added before Figure2.).

Point 7:

Page 4, Lines 166 and 168: The drift ratio θ is not reported in the equation (2.1);

Response 7:

The revised portion are as follows:      (in Section 2)

Point 8:

Page 6, Lines 211 and 213: “the actual strain σ”; it is probably “the actual stress σ”;

Response 8:

The revised portion are as follows: “…the actual stress σ…”

Point 9:

Page 6, Lines 233 and 234: Move the following sentence “The detailed geometric parameters are listed in Table1, Table 2 and Figure 8” to the beginning of the paragraph. Anticipate figure 8 by numbering it with number 5. Then arrange the numbering of the following figures.

Response 9:

The original Figure 8 had been numbered with Figure7 and the following figures were readjusted. The sentence “The detailed geometric parameters are listed in Table1, Table 2 and Figure 7”had been moved to the beginning of the paragraph.

Point 10:

Page 8, Lines 289, 290: Correct the numbering in Figure 8.

Response 10:

We had corrected the numbering in Figure 7(originalFigure8)

Point 11:

Page 7, Line 254: “The”; it is better “the”.

Response 11:

We had revised “The HESs” as “the HESs”

Round 2

Reviewer 2 Report

the authors have complied with my comments and suggestions. Therefore, please accept this manuscript, in its current form, for publication.

Author Response

We thank for the positive comments.

Please see the attached files.
